# “I Shouldn’t Have to Drive to the Suburbs”: Grocery Store Access, Transportation, and Food Security in Detroit During the COVID-19 Pandemic

**DOI:** 10.3390/nu17152441

**Published:** 2025-07-26

**Authors:** Aeneas O. Koosis, Alex B. Hill, Megan Whaley, Alyssa W. Beavers

**Affiliations:** 1Department of Nutrition and Food Science, Wayne State University, Science Hall 410 W Warren, Detroit, MI 48201, USA; 2Department of Urban Studies and Planning, Wayne State University, 3198 Faculty/Administration Building (FAB) 656 W. Kirby St., Detroit, MI 48202, USA; 3College of Nursing, Michigan State University, 1355 Bogue Street, Suite 120, East Lansing, MI 48824, USA; whaleym1@msu.edu

**Keywords:** food insecurity, food access, food environment, grocery store, food retail, transportation, COVID-19 pandemic

## Abstract

Objective: To explore the relationship between type of grocery store used (chain vs. independent), transportation access, food insecurity, and fruit and vegetable intake in Detroit, Michigan, USA, during the COVID-19 pandemic. Design: A cross-sectional online survey was conducted from December 2021 to May 2022. Setting: Detroit, Michigan. Participants: 656 Detroit residents aged 18 and older. Results: Bivariate analyses showed that chain grocery store shoppers reported significantly greater fruit and vegetable intake (2.42 vs. 2.14 times/day for independent grocery store shoppers, *p* < 0.001) and lower rates of food insecurity compared to independent store shoppers (45.9% vs. 65.3% for independent grocery store shoppers, *p* < 0.001). Fewer independent store shoppers used their own vehicle (52.9% vs. 76.2% for chain store shoppers, *p* < 0.001). After adjusting for socioeconomic and demographic variables transportation access was strongly associated with increased odds of shopping at chain stores (OR = 1.89, 95% CI [1.21,2.95], *p* = 0.005) but food insecurity was no longer associated with grocery store type. Shopping at chain grocery stores was associated with higher fruit and vegetable intake after adjusting for covariates (1.18 times more per day, *p* = 0.042). Qualitative responses highlighted systemic barriers, including poor food quality, high costs, and limited transportation options, exacerbating food access inequities. Conclusions: These disparities underscore the need for targeted interventions to improve transportation options and support food security in vulnerable populations, particularly in urban areas like Detroit. Addressing these structural challenges is essential for reducing food insecurity and promoting equitable access to nutritious foods.

## 1. Introduction

Food insecurity is a household-level economic and social condition of limited or uncertain access to adequate food [1]. In 2023, approximately 13.5% of American households experienced food insecurity, a significant increase from the pre-pandemic rate of 10.5% in 2019, underscoring the impact of the COVID-19 pandemic and its disruptions [1]. The pandemic’s economic shocks, including job losses, supply chain disruptions, and inflation, disproportionately affected communities already marginalized by structural racism, amplifying pre-existing food access inequities [2,3]. Food insecurity disproportionately affects low-income households (37% experience food insecurity), female-headed households with no spouse (35%), and Black (23%) and Hispanic (22%) households [1]. Adults experiencing food insecurity are more likely to have inadequate intakes of nutrients such as calcium, iron, protein, and fiber, due to reduced intake of nutrient-rich food groups like dairy, fruits, and vegetables [4]. Additionally, food insecurity is associated with higher rates of diabetes and elevated cardiovascular risk [5]. As such, food insecurity is recognized as a social determinant of health (SDOH), which the Centers for Disease Control and Prevention define as “nonmedical factors that influence health outcomes” [6]. Diet-related chronic diseases contribute significantly to the burden of many of the leading causes of death in the United States. Racial disparities in diet-related chronic diseases are well-documented, with Black Americans often suffering the highest rates of illness and death [7].

Health risks and resources are both spatially and socially structured, and Black Americans disproportionately reside in economically disadvantaged neighborhoods. There is a growing focus on how these residential environments impact health and contribute to racial health disparities. In fact, the SDOH framework includes neighborhood and built environment as one of the five domains that shape health [8]. Historically excluded communities, including Black, Latino, and Native American, have been subjected to discriminatory housing practices such as redlining, segregation, and exclusion from wealth-building opportunities like homeownership [9]. Research has shown that living in economically disadvantaged neighborhoods is associated with a range of negative diet-related health outcomes, even when accounting for individual socioeconomic status [10].

Historical practices of structural racism have significantly shaped racial and economic disparities in urban areas, including food access inequities [11]. Redlining, a mid-20th-century discriminatory practice, systematically denied mortgage lending and insurance to neighborhoods based on their racial or ethnic composition, as outlined in red as “hazardous” areas by the Home Owners’ Loan Corporation maps [9]. This led to disinvestment in predominantly Black and immigrant neighborhoods, influencing racial segregation, housing quality, and economic opportunities [12]. Although officially banned in 1968, redlining’s influence is still evident today, including in the geographic distribution of grocery stores and other essential services. Previous research has shown that neighborhoods that were redlined often have fewer grocery stores, especially those offering a wide selection of fresh and healthy foods [12,13].

While not solely attributable to redlining, Black and low-income neighborhoods typically have a greater number of lower quality food outlets and fast food outlets [14]. Across the United States, chain supermarkets were found to be 52% less available in Black ZIP codes and 32% less available in Hispanic ZIP codes compared to White ZIP codes, even after controlling for income [15]. Additionally, inner cities with higher rates of poverty, unemployment, and vacant units tend to have limited access to larger supermarkets, which are more commonly found in suburban and wealthier areas [12]. Instead, these inner cities are often served by independent and small grocery stores, discount stores, and corner stores [16]. Studies have found that large-scale chain supermarkets are associated with increased fruit and vegetable consumption across various ethnic groups [17]. Neighborhoods with fewer supermarkets often have a higher prevalence of small convenience stores and fast-food outlets, which provide limited healthy food options [18]. This limited access to supermarkets may restrict the availability of healthy foods, potentially contributing to higher rates of heart disease, and diabetes among racial and ethnic minorities [19]. However, findings about the association between local food access and food insecurity have been inconsistent, with results varying based on community makeup and food access measurements [17]. Recent discourse about these racial disparities in food access favors the term “food apartheid” over “food desert” to emphasize the systemic and deliberate policy decisions behind these disparities. Unlike “food desert,” which can imply a naturally occurring phenomenon, “food apartheid” frames the issue as a human-created system of segregation which led to an inadequate, inequitable, and unjust food environment [20]. This framework aligns with conceptualizations of structural racism as “the totality of ways, in which societies foster racial discrimination, via mutually reinforcing inequitable systems” including housing, employment, and other sectors that shape neighborhood environments [21]. The built environment becomes a key mechanism through which these mutually reinforcing systems operate, as zoning policies, transportation infrastructure, and retail development decisions systematically determine which communities have access to quality food retail and which do not [22,23].

Detroit, MI, once the fifth-largest metropolis in America, is estimated to have a food insecurity rate of 31%, and 69% of Detroit households struggle to cover basic needs, including food [24,25]. Detroit has experienced decades of deindustrialization, population decline, and the marginalization of its majority-Black population [26]. This post-industrial decline is not unique to Detroit; other cities in the United States have experienced similar occurrences [13]. In Detroit, these changes have shaped the local food environment. The city lacks a substantial number of large chain supermarkets, directing residents towards full-line independent grocery stores and gas stations. Detroit’s food environment is emblematic of broader economic conditions and business decisions that have contributed to economic inequalities and racial segregation, resulting in disparate impacts on food access in Black neighborhoods [9,13].

Building upon these longstanding inequities, the COVID-19 pandemic forced shifts in eating behaviors and food purchasing patterns [27]. The pandemic revealed how structural racism creates cascading vulnerabilities: neighborhoods shaped by decades of discriminatory policies faced compounded challenges as small independent stores struggled with supply chain disruptions, large chains reduced hours or closed locations in “unprofitable” areas, and public transportation became a health risk [28,29]. Economic pressures from supply chain disruptions and inflation led to changes in household spending, favoring purchasing groceries over dining out [27,30]. Vulnerable populations, who were already at risk for food insecurity, experienced heightened challenges as they navigated these changing circumstances [2]. The pandemic disrupted access to food, particularly in urban environments like Detroit, where residents already faced limited access to supermarkets and fresh, affordable food options [31]. The United States Bureau of Labor Statistics reported above average price increases in the Detroit-Warren-Dearborn metropolitan area for all months and for nearly all items during the COVID-19 pandemic [32,33,34,35]. The combination of a unique urban food environment, historical disinvestment, and pandemic-driven disruptions likely intensified disparities in food access, making it essential to examine the full extent of these changes in dietary quality and food security levels.

While research has examined food environments and shopping patterns, limited studies have investigated the association between shopping at independent versus chain grocery stores and food security, particularly in economically disadvantaged urban settings. This gap is especially relevant in Detroit, where the COVID-19 pandemic may have disproportionately impacted these retail formats. The pandemic’s impact on transportation patterns, particularly reduced public transit service and increased reliance on personal vehicles, may have differentially affected access to chain versus independent stores.

Our study investigates how food insecurity intersects with multiple SDOH domains, particularly through economic stability constraints (food insecurity) and neighborhood and built environment factors (food retail access, transportation usage). Understanding these interconnections helps contextualize how transportation barriers, store accessibility, and socioeconomic factors work together to create systematic disadvantages in food access. This study’s primary objective is to investigate the associations between grocery store type (independent vs. chain) an individual typically shops at, transportation access, food security status, and fruit and vegetable intake in Detroit during the COVID-19 pandemic. Our hypothesis is that residents of Detroit who use their own vehicle and shop at chain grocery stores will exhibit lower levels of food insecurity and higher fruit and vegetable intake compared to those who shop at independent grocery stores and do not use their own vehicle for grocery shopping.

## 2. Methods

Detroit, Michigan, with a population of approximately 640,000 is a majority-Black city (78%) with a poverty rate of 32% [36]. Following decades of deindustrialization and population decline, Detroit has a limited supermarket chain presence, with residents primarily served by independent grocery stores. Our original sampling goal was to obtain a geographically representative sample across Detroit’s 54 Master Plan Neighborhoods, with stratification ensuring coverage of neighborhoods varying in socioeconomic status and food access. Data collection occurred between December 2021 and May 2022. The first stage of recruitment for the survey used flyers mailed to 10,000 Detroit households that were selected using a stratified random sample within the city of Detroit’s 54 Master Plan Neighborhoods. Flyers were mailed to each of these households three times. Flyers were also provided to local food assistance organizations to distribute to their clients. Due to the low flyer response, *n* = 219, recruitment was expanded to include Facebook advertisements targeted at Detroit residents and yielded 437 responses. We were not able to calculate survey response rate because neither the number of flyers distributed by food assistance organizations nor the Facebook ad impressions were tracked. Inclusion criteria were: (1) age 18 or older, and (2) current Detroit residency (self-reported). Exclusion criteria included inability to complete the survey in English and residing outside Detroit city limits. The surveys were completed online. Quality screening involved removing responses with duplicate timestamps, impossibly fast completion times (<50% of median), nonsensical open-text entries, and incorrect ZIP code [37]. After excluding incomplete or inconsistent responses, the final analytical sample included 656 participants. Participants were incentivized to participate with a chance to win one of 100 $25 gift cards.

Demographic data collected included age, gender, race/ethnicity, education level, and income [3]. The income-to-poverty ratio (IPR) was calculated by dividing the mid-point of each income bracket by the federal poverty line for the corresponding household size [38]. Food security was assessed using the USDA’s six-item short-form household food security module and classified into high, low, and very low [39]. Participants were asked the name and city of the main grocery store from which they obtained food, and the mode of transportation used to obtain food [3]. The researchers then categorized grocery stores by their location (inside or outside the city of Detroit) and type (chain or independent). For this study, a chain grocer was defined as any full-line grocery store chain that operates under a corporate structure, including standard full-line grocery stores (e.g., Kroger), supercenters (e.g., Wal-Mart), discount chains (e.g., Aldi), and wholesale clubs (e.g., Costco). Independent grocers were defined as having one or a small number of locally owned and operated full-line grocery store locations. This classification was based on a preexisting classification of Detroit full-line grocery stores [40], with any ambiguous cases resolved by investigative team consensus. The frequency of fruit and vegetable consumption was assessed using The National Cancer Institute’s Dietary Screener Questionnaire [41]. To better understand participants’ lived experiences and allow for more personal responses, optional open-ended questions were included. Participants were asked, ‘What, if anything, would help your household to meet its food needs?’ and ‘Do you have any additional comments or experiences about food you want to share?’ [3].

### Analysis

All analyses were performed using Stata version 16.0 (StataCorp LLC, College Station, TX, USA). Bivariate associations between grocery store type and transportation methods, fruit and vegetable intake, store location, and demographic characteristics were assessed using chi-square tests for categorical variables and Wilcoxon rank-sum tests for ordinal or non-normally distributed continuous variables. To examine factors associated with grocery store type, logistic regression models were used. The primary outcome was grocery type (chain versus independent stores). Control variables were selected based on reviewing relevant literature [1,16,42]. Four sequential models were specified to assess the robustness of associations while adjusting for potential confounders: Model 1 (unadjusted) included only the primary predictor variable; Model 2 adjusted for age and gender; Model 3 further adjusted for race and income-to-poverty ratio and Model 4 further adjusted for SNAP participation. Results are presented as odds ratios (OR) with 95% confidence intervals (CI). An OR >1 indicates higher odds of shopping at chain stores compared to independent stores. Statistical significance was set at *p* < 0.05. Multiple linear regression analyses were conducted to examine associations between grocery type and log of fruit and vegetable intake frequency, with similar covariate adjustment strategies as described above. Fruit and vegetable intake was log-transformed due to non-normal distribution of residuals. Coefficients were then back-transformed to times per day to provide interpretable effect sizes. To assess the presence of multicollinearity, variance inflation factors (VIF) for all variables were determined. Variance inflation factors for these models are found in Appendix A. All values were below conventional thresholds (all VIF < 10), indicating no problematic multicollinearity [43].

We conducted qualitative analysis of participants’ responses to the open-ended questions examining perceptions of their local food environment and what would help them meet their food needs. Two researchers independently reviewed responses to develop a preliminary codebook, identifying themes related to food access experiences. The codebook was developed inductively through iterative refinement: (1) independent preliminary coding of responses, (2) comparison and discussion to develop unified codes, (3) refinement of code definitions, and (4) systematic application to all responses. Any coding discrepancies were resolved through consensus discussion involving the two coders and principal investigator until agreement was reached for each response [44,45]. Prior to consensus 63% of responses were coded identically, an additional 20% had at least one code applied by both coders, 17% had no overlapping codes. Twenty-seven codes were ultimately developed. We retained themes describing food quality, variety, and affordability; chain versus independent store presence; transportation and geographic barriers; and structural factors influencing these conditions, as these related directly to our quantitative research questions.

## 3. Results

The survey respondents were predominantly non-Hispanic Black (61%), followed by non-Hispanic White (23%), non-Hispanic Other (10%), and Hispanic (6%) (Table 1). About 40% of participants had some college or an associate’s degree, and 34% were college graduates or higher. Most respondents were female (79%), and the average age was about 51 years. Just under half, 49%, were food insecure, with 22% experiencing low food security and 29% very low food security. When comparing chain store shoppers with independent store shoppers, gender and age were similar, but there were significant differences in all other characteristics examined. Notably, food security status differed significantly (*p* < 0.001). Among chain store shoppers, 54.1% reported high food security, compared to 34.6% of independent store shoppers. The prevalence of low food security was similar between the two groups (21.7% for chain store shoppers and 23.5% for independent store shoppers), but very low food security was more prevalent among independent store shoppers (41.8%) than chain store shoppers (24.2%). Income-to-poverty ratio differed significantly (*p* < 0.001), with chain store shoppers having a higher mean IPR (2.19 ± 1.43) compared to independent store shoppers (1.34 ± 0.94). A greater proportion of chain store shoppers had at least some college education (81.2%) compared to independent shoppers (53.4%) (*p* < 0.001). Additionally, larger households (5 or more members) were more common among independent store shoppers (33.7%) compared to chain store shoppers (20.7%). Our sample differed from Detroit’s general population in several key demographics (Appendix A). The sample was disproportionately female (78.6% vs. 53.5%) and older (21.6% aged 65+ vs. 18.5%). Our sample was more highly educated (33.4% with bachelor’s degree or higher vs. 15.1%), but lower income; for example 39.6% of our sample had household income under $20,000 annually vs. 33.2% for the city of Detroit. Our sample had a lower proportion of Black residents compared to Detroit overall (60.6% vs. 76.6%), more households with seniors (38.6% vs. 27.4%), and more households with children (34.6% vs. 28.4%), suggesting potential overrepresentation of multigenerational households [46].

Table 2 presents transportation method, grocery location, and fruit and vegetable intake by grocery store type. Significant differences were observed in transportation methods between the two groups (*p* < 0.001), with 52.9% of independent store shoppers using their own vehicle, compared with 76.2% of chain store shoppers. The location of the main grocery store also differed significantly between the groups (*p* < 0.001), with 82.5% of independent store shoppers conducting their shopping inside Detroit, compared to 39.3% of chain store shoppers. The fruit and vegetable intake frequency was significantly different between the groups (*p* < 0.001), with independent store shoppers reporting a mean fruit and vegetable consumption of 2.14 ± 1.93 times per day and chain store shoppers reporting a mean of 2.42 ± 1.74 times per day. A significantly higher intake frequency was found among chain store shoppers for fruit, leafy vegetables, and other vegetables. There was no significant difference between groups in potato or juice consumption.

Table 3 presents the results of logistic regression models examining the associations between grocery store type, method of transportation, location of grocery store, and food security status, with odds ratios (OR) representing the likelihood of shopping at chain grocers versus independent stores. For each research question, four models were analyzed: an unadjusted model (Model 1); a model adjusted for age and gender (Model 2); a model including age, gender, race, and income-to-poverty ratio (Model 3); and a fully adjusted model including age, gender, race, income-to-poverty ratio and SNAP participation (Model 4). Across all four models, transportation was a significant predictor of grocery store type, with individuals using a personal vehicle having higher odds of shopping at chain stores (Model 4: OR = 1.89, 95% CI 1.21–2.95, *p* = 0.005) compared with those who did not use their own vehicle to obtain food. Primarily shopping at a grocery store located in Detroit was associated with lower odds of shopping at chain stores across all models (Model 4: OR = 0.13, 95% CI [0.08,0.21], *p* < 0.001) compared to those shopping outside of Detroit. The association between food insecurity and grocery store type varied across models and levels of food insecurity. In Model 4, the association between grocery store type and low food security (OR = 1.10, 95% CI: 0.64–1.91, *p* = 0.729) and very low food security (OR = 0.65, 95% CI: 0.39–1.07, *p* = 0.09) was no longer significantly different than those with high food insecurity. Additionally, linear regression models assessed the relationship between grocery store type and fruit and vegetable intake. Coefficients were back-transformed from the log scale to provide interpretable effect sizes. Chain store shoppers had significantly higher fruit and vegetable intake compared to independent store shoppers across all models: Model 1 1.28 times per day, (95% CI: 1.11, 1.48, *p* < 0.001); Model 2 1.26 times per day, 95% CI: 1.09, 1.46, *p* = 0.002); Model 3 1.18 times per day, 95% CI: 1.01, 1.38, *p* = 0.037); and Model 4 (1.18 times per day, 95% CI: 1.01, 1.38, *p* = 0.042).

Participants were also asked open-ended questions about food accessibility, affordability, and the grocery environment in Detroit. These questions asked for additional comments and suggestions for what else would help them meet their food needs. Responses revealed three main themes: food quality and availability, food accessibility, and food affordability.

### 3.1. Food Quality and Availability

Concerns about the quality and availability of food were frequently mentioned by participants (*n* = 80). This dissatisfaction was most commonly directed at the selection and freshness/quality of produce and meat in Detroit independent stores. As one respondent noted, “I do not shop at grocery stores in Detroit because the[y] are mostly neighborhood grocery stores and the quality of the fresh fruits and vegetables, and meat is poor compared to the suburban grocery stores and grocery store chains.” As this quote illustrates, participants explicitly described inferior food quality within Detroit stores compared with the suburban stores, and/or inferior quality in independent grocery stores compared with chain stores. In Michigan, major grocery chains include Meijer, Kroger, and Aldi. Within Detroit, however, the grocery landscape is marked primarily by independent grocers and neighborhood markets. However, some even perceived chain stores in Detroit as falling short, with one participant stating, “The [Location and Store Name] is both poorly stocked and has a poor selection of healthy food.” Some participants highlighted these inequities in food access, with one participant saying, “Detroit is a food desert for the Black community” and another saying “My main issue is the quality and choices of food available in low-income areas. I shouldn’t have to drive to the suburbs to have access to fresh fruit and vegetables. … the local stores should be fined for what they sell…. We deserve better.” In response to what would help to better meet their food needs, responses under this theme included increasing the availability of high quality, fresh, and/or healthy foods. As one respondent expressed, “I would like to see fresh foods and non-outdated, about-to-go-bad foods here in the city.”

### 3.2. Food Accessibility

Food accessibility emerged as a key theme, including both the ability to obtain food through transportation or delivery, and the geographic availability of grocery stores within Detroit. Transportation emerged as a significant barrier to accessing food, with 59 participants emphasizing the challenges faced by themselves or others without reliable vehicles or lack of proximity to preferred stores. One respondent shared, “I rely completely on car transportation to get affordable, good-quality groceries,” while another described the struggles of being without a car: “While the four months my car was broken, there was no transportation aid program.” Many participants expressed frustration over the need to leave their neighborhood or the city to access quality grocery stores. Some respondents desired more food retailers within neighborhoods: “Need more small markets/quality grocery stores near the neighborhoods on the west side.” Others highlighted the distance and limited accessibility of existing stores, with one participant noting, “The markets are spaced too far apart; the market near me is 7 blocks away.” Some participants expressed a desire for more chain stores within Detroit and called for improved stocking to ensure equitable access to fresh food. One participant noted that they would like to see, “More chain stores located in Detroit that are stocked the same way they are in the suburbs,” highlighting the desire for trusted grocery chains to expand their presence. Other proposed solutions to improve accessibility included expanding public transit options, providing free or low-cost delivery services, and offering rides to grocery stores. This theme underscores the uneven geographic distribution of food retailers across Detroit, disproportionately affecting some residents’ ability to access a nutritious diet.

### 3.3. Food Affordability

The high cost of food was a central challenge for many participants (*n* = 70). Rising prices, particularly for fresh produce, were frequently cited as barriers to purchasing nutritious items. One respondent noted, “The cost of food is too high. Therefore, some foods I refuse to purchase,” while another added, “Fruits and vegetables are twice as expensive.” While most people who mentioned high prices did not specify location or type of, store, some respondents explicitly described higher prices at the independent stores compared with chains. For example, one participant stated, “A box of cereal at Target is $3.50 for Honey Nut Cheerios and $6.00 for the same size box at the Chaldean-owned stores.” These affordability challenges were further compounded by perceptions of price disparities within Detroit, where urban residents often felt they were paying more for less. As one participant explained, “The neighborhood grocery stores are always more expensive than Kroger or Meijer and have lower quality food.” Such examples reflect the broader perception of inequities in food pricing, where urban residents face higher costs for lower-quality products, exacerbating food insecurity for many. Additionally, participants with specific dietary needs (*n* = 16) highlighted challenges in finding reasonably priced foods. One respondent explained, “There are food sensitivities in our family, and that makes it even harder to find reasonably priced food.” Recommendations included increasing the availability of affordable gluten-free and vegetarian options, with specific suggestions such as “healthy gluten-free options that are affordable” and “more sales for vegan/vegetarian products.” To address these issues, participants suggested lowering food prices or providing financial assistance to get food such as gas cards, covering delivery fees, and free transportation.

The themes of food quality, accessibility, and affordability are deeply interconnected in participants’ responses. One participant captured this overlap, stating, “Not having a car during the COVID pandemic made food extremely expensive. The neighborhood grocery stores are always more expensive than Kroger or Meijer and have lower quality food.” Similarly, another shared, “The produce at stores near me is not always fresh. It’s also expensive and hard to access without a car.” These quotes illustrate how transportation barriers can limit food retail options to local retailers, which are higher cost and sometimes have poor food quality.

## 4. Discussion

This study examined the relationship between grocery store type, food insecurity, and fruit and vegetable intake among Detroit residents during the COVID-19 pandemic. Chain store shoppers were more likely to report higher food security, higher income-to-poverty ratios, and greater educational attainment compared to independent store shoppers, who were more likely to live in larger households and shop primarily within Detroit. We found that individuals using personal vehicles and those shopping outside of Detroit had higher odds of shopping at chain stores. Very low food security was associated with a lower likelihood of shopping at chain stores in unadjusted and partially adjusted analyses. However, this association diminished after accounting for race and socioeconomic factors, suggesting that these variables may influence the relationship between food security and shopping patterns. Individuals who shopped primarily at independent stores had lower fruit and vegetable intake compared to chain store shoppers. While our binary store categorization enabled analysis of broad retail patterns, it obscures important heterogeneity within store types, including ethnic markets, food cooperatives, and varying price points that may independently influence food access and dietary outcomes among diverse populations. Additionally, in qualitative responses participants expressed concerns about food quality, affordability, and access, revealing systemic barriers such as the high cost of food, lack of quality grocery stores within Detroit, and the challenges faced by residents relying on public or shared transportation.

The study period coincided with unprecedented food price inflation, making Detroit a particularly critical location for examining food shopping behaviors. The pandemic related global market disturbances resulted in the largest and most rapid increase in food inflation in the United States in over 40 years [47]. The Detroit-Warren-Dearborn metropolitan statistical area experienced disproportionately severe food price increases, with the consumer price index for foods consumed at home rising 10.0% in December 2021, 11.4% in February 2022, 14.3% in April 2022, and 16.6% in June 2022, consistently exceeding national averages by approximately 3% [32,33,34,35]. This inflationary environment likely intensified existing economic pressures on food-insecure households, potentially altering shopping venue choices and highlighting the importance of understanding how different retail environments serve vulnerable populations during periods of economic stress.

The cross-sectional nature of this study limits our ability to establish causal relationships or determine the direction of associations between food security status and grocery store type. Looking at the SDOH, it is plausible that food insecurity influences store choice through economic stability constraints that limit options to nearby independent stores, or conversely, that shopping at stores with limited healthy food options contributes to food insecurity over time through the neighborhood and built environment domain. These relationships likely involve bidirectional influences and unmeasured confounding factors across multiple SDOH domains. The complex interplay between transportation access, economic resources, neighborhood food environment, and food security creates a web of mutually reinforcing disadvantages spanning the economic stability, built environment, and social context domains that cannot be disentangled with cross-sectional data.

While there is limited existing research examining chain compared with independent grocery stores, research consistently demonstrates an association between the presence of supermarkets and lower obesity rates [17]. Conversely, smaller stores, which often dedicate more shelf space to energy-dense, nutrient-poor foods, are associated with higher obesity rates, even after controlling for walkability, store size and individual behavioral factors [18,48,49,50]. Data from the Food Acquisition and Purchasing Study (FoodAPS) reveals that food acquisitions from supermarkets have significantly higher Healthy Eating Index (HEI-2010) scores than other store types, roughly 10 points higher than acquisitions from small and specialty food stores, and between 11 and 17 points higher than acquisitions from convenience, dollar, and other grocery stores [51]. Additionally, smaller stores often offer damaged or unappealing fruits and vegetables, which discourages purchases and impacts dietary intake [52]. Detroit residents often view the food environment as inequitable, with the prevalence of lower-quality stores reflecting broader systemic barriers that restrict access to nutritious and affordable food [11,53]. Recent qualitative studies provide insights into individuals’ experiences with their local food environments. For example, a study in Hartford, Connecticut, found that residents in low-income neighborhoods frequently voiced dissatisfaction with the quality and variety of fresh produce available in local stores, noting that smaller stores often stocked damaged or unappealing produce [52]. Similarly, research in Philadelphia revealed that residents often chose to shop outside their neighborhoods to make bulk purchases, they felt compelled to shop locally when no other viable options were available [54]. This pattern of shopping underscores the constraints imposed by systemic inequities in food access, where local food options may be inadequate or overpriced, forcing residents to develop strategies to meet their food needs.

We found that transportation mode was associated with grocery store type, with respondents using a personal vehicle to obtain food having a higher odds of shopping at chain grocers. This quantitative pattern was echoed in qualitative responses, where participants described the necessity of traveling outside Detroit for quality food options, often requiring personal vehicles due to inadequate public transit. According to the 2023 American Community Survey, it is estimated that 28% of Detroiters do not have a vehicle available, a higher share than many comparable cities (e.g., 8% in Baltimore MD, 22% in Cleveland OH, 12% in Milwaukee WI, 10% in Memphis TN, and 29% in Philadelphia PA) [46]. Adequate transportation is associated with access to healthy foods and is associated with lower food insecurity, while food insecurity is associated with greater reliance on small food stores, corner stores, and traveling fewer miles for shopping [55,56]. Participants revealed how transportation constraints force reliance on local independent stores despite preferences for chain alternatives, with one noting the challenge of “getting to better stores when you don’t have a car and the bus doesn’t go there reliably”. Several studies have shown that individuals with lower incomes often travel beyond their neighborhoods, relying on personal vehicles or ride-sharing services to access stores with a wider food selection, and to access higher-quality, healthier, or more affordable food options [57,58]. In contrast, individuals without vehicles face significant limitations, relying on limited public transportation networks or walking to the nearest available store. In Detroit, public transit inefficiencies and limited coverage exacerbate these challenges, making access to distant grocery stores particularly difficult. Studies consistently rank Detroit’s public transit system among the least effective in the United States, with limited bus routes and infrequent schedules creating significant barriers for low-income residents [56]. The COVID-19 pandemic intensified these transportation challenges, as public transit usage plummeted in Detroit and other urban centers due to service reductions and health concerns, disproportionately affecting lower-income populations [28]. The increased travel burdens for food access have been documented not only in Detroit but also in cities like Seattle, WA, Portland, OR, and Jackson, MS [59,60,61]. Similarly, qualitative research in Pennsylvania underscores the difficulties faced by low-income adults, including reliance on inefficient public transportation and the need to walk long distances to grocery stores [62].

These transportation barriers are a critical aspect of systemic inequalities in food access, rooted in historical patterns of disinvestment and segregation [9]. The transportation and geographic barriers identified align with the SDOH neighborhood and built environment domain, where limited public transit and spatial distribution of food retailers create systematic disadvantages. Respondents reported poor quality and limited choices in low-income areas, often necessitating trips to multiple stores or to suburbs for fresh produce. These findings align with previous research in urban settings, which has documented patterns of disparate access to nutritious and affordable food options among lower-income populations [5,12,50]. Limited vehicle access, in particular, has been strongly linked to food insecurity among lower-income populations, with reliance on public transportation, ride-sharing, or other alternatives further compounding the issue [54,56,63]. Transportation barriers remain a well-documented contributor to higher rates of food insecurity, underscoring the complex relationship between transportation, food access, and food security in urban environments [56].

Improving food access in underserved urban areas requires a multifaceted approach that addresses both immediate barriers and systemic inequities through evidence-based interventions. Incentive programs like Double Up Food Bucks, which matches SNAP benefits dollar-for-dollar when purchasing fruits and vegetables at participating retailers, have demonstrated effectiveness in increasing fruit and vegetable consumption among low-income households [64,65,66]. Existing efforts to address food access in urban areas include local initiatives like the Green Grocer Project in Detroit, which works with local grocers to stock fresh produce and engages the community to align offerings with local needs [67]. Similar programs have been implemented in other cities, including Healthy Retail SF in San Francisco, CA, which transforms corner stores into outlets for fresh, nutritious foods through technical assistance and financial support, and the Healthy Corner Store Initiative in Philadelphia, PA, which partners with small retailers to increase access to fruits, vegetables, and other healthy options while providing nutrition education [68,69]. These approaches aim to improve both the availability and affordability of healthy foods in small retailers while addressing economic barriers to nutritious food access. Additionally, supporting small and independent retailers through incentives and technical assistance could help expand access to fresh, nutritious foods, while improving public transportation networks and subsidized delivery services may reduce the mobility challenges faced by those without personal vehicles. Efforts to enhance the sustainability of small retailers could include cooperative purchasing arrangements, such as those used by the Rural Access Distribution (RAD) Cooperative in North Dakota, which reduce costs and expand product offerings through shared resources, alongside mobile markets and flexible retail solutions that bring fresh, affordable food directly to communities with limited access [70]. Public policies that provide grants or tax relief for small retailers upgrading infrastructure, as exemplified by California’s Healthy Refrigeration Grant Program, can incentivize investment in healthier food offerings [70]. Addressing the systemic nature of food inequities requires innovative, community-driven solutions. A sign of progress is the opening of Detroit People’s Food Co-op, the first Black-led and operated full-line grocery store in Detroit in decades, a community-led approach to improve food access and reduce disparities.

Equally important is addressing the lasting impact of historical injustices such as redlining, which have shaped the distribution of food resources in urban environments. Interventions should prioritize historically underserved areas, incorporating the legacy of systemic inequities into urban planning and food policy. Aligning food assistance programs, such as SNAP and Special Supplemental Nutrition Program for Women, Infants, and Children (WIC), with local food systems—through increased benefits, broadened eligibility, and support for delivery services—can further enhance their effectiveness. Finally, fostering community engagement and local ownership in these efforts ensures that interventions are responsive to the specific needs and priorities of the population, creating equitable and sustainable food systems that promote long-term health and well-being.

## 5. Conclusions

This study demonstrates existing inequities in grocery store access, transportation, and food security in Detroit, particularly during the COVID-19 pandemic. Our findings suggest that food insecurity in urban areas like Detroit is intricately related to structural inequalities, including disparities in transportation access and the availability of high-quality food sources. The differences observed between chain and independent grocery store shoppers highlight broader divides that continue to shape Detroit’s food environment.

Persistent barriers to accessing fresh, healthy foods in low-income and minority neighborhoods demand targeted interventions that address both immediate community needs, and the structural factors associated with food insecurity. Comprehensive strategies are needed to enhance transportation infrastructure, increase the availability of high-quality food outlets in underserved areas, and support local retailers in providing healthier options. Recognizing and addressing the historical legacy of systemic injustices, such as redlining, is essential for creating a more equitable food system that ensures all residents have access to nutritious and affordable food. Future research is needed to develop and implement tailored, locally informed approaches to improving food access. These approaches should consider the social, economic, and cultural contexts of individual communities to address the multifaceted nature of food insecurity.

### 5.1. Strengths

This study has several strengths that enhance its contribution to the literature on food insecurity and food access disparities in urban settings. By focusing on Detroit, a historically underrepresented population disproportionately affected by systemic inequities, the research provides critical insights into how structural barriers, such as transportation and grocery store type, intersect with food security and dietary outcomes. The mixed-methods approach, integrating quantitative survey data with qualitative responses, offers a comprehensive understanding of participants’ lived experiences. Conducted during the COVID-19 pandemic, the study captures the compounded effects of pre-existing disparities and pandemic-related disruptions. Finally, the results are framed within Detroit’s unique food environment, contextualizing the findings for broader discussions on urban food systems and informing targeted interventions to address food insecurity in vulnerable populations.

### 5.2. Limitations

This study has several limitations that should be considered when interpreting the results. The cross-sectional design does not permit examining causal relationships between grocery store type and transportation, fruit and vegetable intake, and food insecurity. Future studies could use designs such as natural experiments or randomized controlled trials to better understand if there is a causal relationship between these variables and, if so, in which direction. The absence of price data limits our ability to disentangle economic accessibility from other factors related to store choice and food security. Another limitation is the potential for non-response bias, as the response to the mailed flyers was low and the use of social media advertisements may have introduced selection bias by reaching a more technologically engaged subset of the population. This recruitment approach likely excluded the most economically disadvantaged residents, as it has been reported that 13% of adults with household incomes below $30,000 lack access to internet technologies at home [71]. Additionally, the dietary assessment used in this study, the DSQ, has limited sensitivity for dietary variety and portion size, and is known to typically result in an underestimate of fruit and vegetable intake. However, it can capture meaningful directional differences in fruit and vegetable consumption patterns when comparing intake among two groups [72]. Our study focused on grocery shopping behavior and did not examine purchases at other types of food stores such as dollar stores. Future research is needed that uses a more holistic approach to assess food purchasing behaviors and their relationship to food insecurity in urban settings. Lastly, as this study focused on the general population of Detroit, it provides a broad overview of the relationships examined, but does not explore the experiences of population subgroups. Future research is needed that focuses on the experiences of populations that are most affected by food insecurity, such as low-income women with children.

## Figures and Tables

**Table 1 nutrients-17-02441-t001:** Demographic characteristics of respondents.

Characteristic	Independent Store Shoppers		Chain Store Shoppers		Total		*p* Value
	*n*	% or SD	*n*	% or SD	*n*	% or SD	
Race/Ethnicity							<0.001
Non-Hispanic Black	109	67.3%	257	58.9%	366	61.2%	
Non-Hispanic White	17	10.5%	118	27.1%	135	22.6%	
Hispanic	16	10.0%	20	4.6%	36	6.0%	
Non-Hispanic Other	20	12.4%	41	9.4%	61	10.2%	
Education							<0.001
Some High School	24	14.7%	22	5.0%	46	7.7%	
High School Graduate	52	31.9%	60	13.7%	112	18.7%	
Some College or Associate’s	61	37.4%	177	40.5%	238	39.7%	
College Graduate or Above	26	16.0%	178	40.7%	204	34.0%	
Gender							0.289
Male	37	22.6%	80	18.2%	117	19.4%	
Female	126	76.8%	351	80.0%	477	79.1%	
Transgender, Non-Binary, or Other	1	0.6%	8	1.8%	9	1.5%	
Age (Years)	50.1	14.5	51.4	14.0	51.1	14.2	0.308
Income-To-Poverty Ratio Mean ± SD	1.34	0.94	2.19	1.43	1.92	1.36	<0.001
Household Size							<0.001
1	23	13.9%	58	13.2%	81	13.3%	
2	28	16.9%	129	29.3%	157	25.9%	
3	34	20.5%	112	25.4%	146	24.1%	
4	25	15.1%	51	11.6%	76	12.5%	
5	17	10.2%	44	10.0%	61	10.1%	
6	15	9.0%	19	4.3%	34	5.6%	
7 Or More	24	14.5%	28	6.4%	52	8.6%	
SNAP							<0.001
Yes	102	61.5%	170	38.5%	272	44.8%	
No	64	38.5%	271	61.5%	335	55.2%	
WIC							0.107
Yes	15	9.04%	24	5.44%	39	6.43%	
No	151	90.9%	417	94.6%	568	93.6%	
Food Security							<0.001
High	53	34.6%	232	54.1%	285	49.0%	
Low	36	23.5%	93	21.7%	129	22.2%	
Very Low	64	41.8%	104	24.2%	168	28.9%	

Rank sum was used for age and income-to-poverty ratio, all other variables used chi square.

**Table 2 nutrients-17-02441-t002:** Transportation methods, grocery store locations, and fruit and vegetable intake by grocery store type.

	**Independent Store Shoppers (*n*)**	**Independent Store Shoppers (%)**	**Chain Store Shoppers (*n*)**	**Chain Store Shoppers (%)**	***p*-Value**
Transportation Used					<0.001
Other transportation	74	47.1%	101	23.8%	
Own Vehicle	83	52.9%	323	76.2%	
Location of Main Grocery					<0.001
Outside Detroit	29	17.5%	267	60.7%	
Inside Detroit	137	82.5%	173	39.3%	
	**Independent Store Shoppers (Mean)**	**Independent Store Shoppers (S.D.)**	**Chain Store Shoppers (Mean)**	**Chain Store Shoppers (S.D.)**	
Fruit and Vegetable Intake Combined	2.14	1.93	2.42	1.74	<0.001
Fruit Intake	0.47	0.58	0.64	0.59	<0.001
Leafy Vegetable Intake	0.32	0.34	0.48	0.49	<0.001
Other Vegetable Intake	0.45	0.46	0.65	0.53	<0.001
Potato Intake	0.26	0.33	0.25	0.27	0.474
Juice Intake	0.63	1.21	0.39	0.73	0.364

Fruit and vegetable intake (combined and individual categories) are presented in times per day. Chi square tests were used to compare transportation type and grocery location between independent and chain store shoppers, all other comparisons used rank-sum tests. A *p*-value of <0.05 is considered statistically significant.

**Table 3 nutrients-17-02441-t003:** Associations between grocery store type and transportation, grocery location, and food security: logistic regression.

	Model 1 OR	95% CI		*p* Value	Model 2 OR	95% CI		*p* Value	Model 3 OR	95% CI		*p* Value	Model 4 OR	95% CI		*p* Value
Personal vehicle used to obtain food ^1^	2.85	1.94	4.19	<0.001	3.04	2.04	4.52	<0.001	1.96	1.27	3.02	0.002	1.89	1.21	2.95	0.005
Primary grocer in Detroit ^2^	0.14	0.09	0.21	<0.001	0.12	0.08	0.20	<0.001	0.13	0.08	0.20	<0.001	0.13	0.08	0.21	<0.001
Low Food ^3^ Security	0.59	0.36	0.96	0.034	0.70	0.42	1.17	0.177	1.09	0.63	1.89	0.762	1.10	0.64	1.91	0.729
Very Low Food Security ^3^	0.37	0.24	0.57	<0.001	0.39	0.25	0.61	<0.001	0.65	0.39	1.07	0.093	0.65	0.39	1.07	0.092

Model 1 is unadjusted, Model 2 is adjusted for age and gender, Model 3 is adjusted for age, gender, race, and income-to-poverty ratio, and Model 4 is adjusted for age, gender, race, and income-to-poverty ratio and SNAP participation. OR = odds ratio. ^1^ Reference category is any other type of transportation, ^2^ Reference is primary grocer located outside city of Detroit, ^3^ Reference is food secure.

## Data Availability

The data presented in this study are available upon reasonable request from the corresponding author. The data are not publicly available due to privacy or ethical restrictions.

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
