# Peer review of "“I Shouldn’t Have to Drive to the Suburbs”: Grocery Store Access, Transportation, and Food Security in Detroit During the COVID-19 Pandemic"

_nutrients, 2025, doi:10.3390/nu17152441_

Round 1

Reviewer 1 Report

Comments and Suggestions for Authors

This study sought to elucidate the relationship between the type of grocery store utilized, access to transportation, food insecurity, and the consumption of fruits and vegetables in Detroit, Michigan, USA, during the COVID-19 pandemic. A logistic regression was used to examine the survey data collected by the authors. Although the explanation and discussion are logical and clear, several questions remain regarding the estimated results of the logistic regression (see Table 3). The odds ratio for low food security demonstrates a notable change, increasing from 0.59 in Model 1 to 0.70 in Model 2 and 1.09 in Model 3, although the odds ratios did not achieve significance in Models 2 and 3. This finding suggests a potential correlation between the newly incorporated explanatory variables and the incidence of low food security. If this is indeed the case, it raises the issue of multicollinearity, which in turn calls into question the reliability of the odds ratio estimation results. Furthermore, it is necessary to determine whether there is an inverse causal relationship between low food security and the choice of store from which the goods are purchased. In such instances, it may be necessary to introduce at least one instrumental variable to address endogeneity. However, I came across many times that public health studies do not pay much attention to multicollinearity and endogeneity issues; therefore, the above comments may not be serious in these fields.

Author Response

Response: Thank you for your thoughtful methodological insights. Your expertise helped us strengthen our statistical approach and better acknowledge the limitations of our cross-sectional design. 

Author Response:

Reviewer 1

Response: Thank you for your thoughtful methodological insights. Your expertise helped us strengthen our statistical approach and better acknowledge the limitations of our cross-sectional design.

  1. Although the explanation and discussion are logical and clear, several questions remain regarding the estimated results of the logistic regression (see Table 3). The odds ratio for low food security demonstrates a notable change, increasing from 0.59 in Model 1 to 0.70 in Model 2 and 1.09 in Model 3, although the odds ratios did not achieve significance in Models 2 and 3. This finding suggests a potential correlation between the newly incorporated explanatory variables and the incidence of low food security. If this is indeed the case, it raises the issue of multicollinearity, which in turn calls into question the reliability of the odds ratio estimation results.

Response: Thank you for this important methodological concern. The following was added to Lines 214-217; see supplemental materials for the newly added tables: “To assess the presence of multicollinearity, variance inflation factors (VIF) for all variables were determined. Variance inflation factors for these models are found in Supplemental Tables 2-5. All values were below conventional thresholds (all VIF < 10), indicating no problematic multicollinearity.”

  1. Furthermore, it is necessary to determine whether there is an inverse causal relationship between low food security and the choice of store from which the goods are purchased. In such instances, it may be necessary to introduce at least one instrumental variable to address endogeneity. However, I came across many times that public health studies do not pay much attention to multicollinearity and endogeneity issues; therefore, the above comments may not be serious in these fields.

Response: We appreciate this methodological point about potential endogeneity. While instrumental variable approaches would be valuable for causal inference, that is beyond the scope of this paper. As is common in public health, we chose our control variables a priori through reviewing the literature to identify potential confounders and have added this detail in lines 202-203: “Control variables were selected based on reviewing relevant literature [1,16,41]. “We have clarified the limitations section to emphasize that causality should not be inferred from our paper in Lines 577-581: “The cross-sectional design does not permit examining causal relationships between grocery store type and transportation, fruit and vegetable intake, and food insecurity. Future studies could use designs such as natural experiments or randomized controlled trials to better understand if there is a causal relationship between these variables, and if so, in which direction.”

Reviewer 2 Report

Comments and Suggestions for Authors

Thank you for the opportunity to evaluate this submission. The study tackles a highly relevant question—how store type and transportation access intersect with food security and fruit-and-vegetable intake in post-pandemic Detroit—using a mixed-methods design and a comparatively large single-city sample (n = 656). The topic is squarely within Nutrients’ remit and the data set has genuine potential to advance urban-nutrition scholarship; nonetheless, several substantive and presentation-level issues must be resolved before the work can be judged publication-ready.

Abstract
The synopsis communicates the main variables and headline findings but repeatedly implies causality in a cross-sectional context (“use of a personal vehicle increased the odds …”) and omits field dates, response rates, and back-transformed effect sizes. Tightening the language to “was associated with,” adding basic survey logistics, and expressing the dietary coefficient in servings or cup-equivalents would render the abstract both accurate and interpretable.

Methodology
Recruitment flow is opaque: mailed flyers and Facebook ads are mentioned without counts of invitations, impressions, or completion rates, leaving sample representativeness uncertain. Key confounders already collected—SNAP participation, distance to the usual store, a price proxy—are not included in the multivariable models, heightening the risk of omitted-variable bias. The binary “chain vs. independent” exposure lacks reproducible coding; discount chains and franchises may be misclassified, jeopardising construct validity. Finally, the qualitative component reports 27 codes but provides neither a guiding framework nor inter-coder reliability, which weakens the mixed-methods claim.

Introduction
The background is conceptually well grounded but occasionally slides into causal rhetoric and cites pre-pandemic sources without explaining why pandemic-era mobility shifts warrant a fresh investigation. Re-framing statements to emphasise associations and integrating more recent “food apartheid” literature would sharpen the rationale.

Results and Interpretation
Quantitative findings are reported with log-transformed coefficients that cannot be translated into daily intake without back-transformation; odds ratios for food-security status sometimes lack 95 % confidence intervals in text and tables. Qualitative themes appear as a standalone paragraph rather than being woven into the discussion, forfeiting the explanatory power that mixed methods can offer. Effect sizes may be inflated by the missing confounders noted above; re-specifying the models and providing variance-inflation factors would clarify robustness.

In light of these concerns—most critically, exposure misclassification, residual confounding, and causal over-statement—I recommend a Major Revision. A carefully executed revision addressing sampling transparency, model specification, exposure definition, qualitative-methods rigour, and language precision could yield a manuscript of publishable quality.

Author Response

Response: We greatly appreciate your comprehensive and constructive review that addressed both methodological rigor and written presentation. Your detailed feedback on our mixed-methods approach and model specifications significantly improved the manuscript's clarity and scientific validity. 

Author Response:

Reviewer 2

Response: We greatly appreciate your comprehensive and constructive review that addressed both methodological rigor and written presentation. Your detailed feedback on our mixed-methods approach and model specifications significantly improved the manuscript's clarity and scientific validity.

Abstract

  1. The synopsis communicates the main variables and headline findings but repeatedly implies causality in a cross-sectional context (“use of a personal vehicle increased the odds …”) and omits field dates, response rates, and back-transformed effect sizes. Tightening the language to “was associated with,” adding basic survey logistics, and expressing the dietary coefficient in servings or cup-equivalents would render the abstract both accurate and interpretable.

Response: Thank you for identifying this language issue. We reviewed the entire manuscript to identify causal language and modify it as needed. We added field dates to the abstract in line 17: “A cross-sectional online survey was conducted from December 2021 to May 2022.” We provided back-transformed effect sizes for fruit and vegetable intake (1.18 times more per day, line 28). Please note that some of that data in the abstract has changed since we have added a 4th statistical model (see response to point #3 below). Response rates were unavailable, as described in the response to # 2 below.

Methodology

  1. Recruitment flow is opaque: mailed flyers and Facebook ads are mentioned without counts of invitations, impressions, or completion rates, leaving sample representativeness uncertain.

Response: We unfortunately do not have documentation regarding the number of flyers distributed by food assistance organizations, nor do we have metrics for the Facebook ads.  However, we have a few additional details that we added: in line 164 we added that, “Flyers were mailed to each of these households three times,” and we added the number of responses from the mailed flyers vs the ads in line 166-167: “Due to the low flyer response (n = 219) , recruitment was expanded to include Facebook advertisements targeted at Detroit residents, and yielded 437 responses.”

To examine the representativeness of the sample, we have added demographic comparisons between our sample and Detroit census data in line 250-258 and supplemental table 1: “ Our sample differed from Detroit's general population in several key demographics (Supplemental Table 1). The sample was disproportionately female (78.6% vs. 53.5%) and older (21.6% aged 65+ vs. 18.5%). Our sample was more highly educated (33.4% with bachelor's degree or higher vs. 15.1%), but lower income; for example 39.6% of our sample had household income under $20,000 annually vs. 33.2% for the city of Detroit. Our sample had a lower proportion of Black residents compared to Detroit overall (60.6% vs. 76.6%), more households with seniors (38.6% vs. 27.4%), and more households with children (34.6% vs. 28.4%), suggesting potential overrepresentation of multigenerational households.” We have also added a discussion of non-response bias to the limitations, lines 582-588: “Another limitation is the potential for non-response bias, as the response to the mailed flyers was low and the use of social media advertisements may have introduced selection bias by reaching a more technologically engaged subset of the population. This recruitment approach likely excluded the most economically disadvantaged residents, as it has been reported that 13% of adults with household incomes below $30,000 lack access to internet technologies at home.”

  1. Key confounders already collected—SNAP participation, distance to the usual store, a price proxy—are not included in the multivariable models, heightening the risk of omitted-variable bias.

Response: We unfortunately do not have data regarding the distance to the usual store nor a price proxy, we apologize if the manuscript seemed to indicate that we did.  The prior version of the manuscript stated that we asked participants to provide the location of their grocery store, but what was actually asked of participants was the city their main grocery store is located in.  We made that change in line 180: “Participants were asked the name and city of the main grocery store from which they obtained food…” We acknowledge the lack of price data in the limitations in lines 581-582: “The absence of price data limits our ability to disentangle economic accessibility from other factors related to store choice and food security.” We appreciate this suggestion to control for SNAP participation, and we have added this in a 4th model in both the logistic and linear regressions.  See lines 282-310, including table 3.

  1. The binary “chain vs. independent” exposure lacks reproducible coding; discount chains and franchises may be misclassified, jeopardizing construct validity.

Response: Thank you for this clarification request, we have added additional details to make the process more transparent, see (Line 183-189): “For this study, a chain grocer was defined as any full-line grocery store chain that operates under a corporate structure, including standard full-line grocery stores (e.g. Kroger), supercenters (e.g. Wal-Mart), discount chains (e.g. Aldi), and wholesale clubs (e.g. Costco). Independent grocers were defined as having one or a small number of locally owned and operated full-line grocery store locations. This classification was based on a preexisting classification of Detroit full-line grocery stores [39], with any ambiguous cases resolved by investigative team consensus.”

  1. Finally, the qualitative component reports 27 codes but provides neither a guiding framework nor inter-coder reliability, which weakens the mixed-methods claim.

Response: We enhanced our qualitative methods description by adding details on codebook development and the iterative consensus process in Lines 220-230: “Two researchers independently reviewed responses to develop a preliminary codebook, identifying themes related to food access experiences. The codebook was developed inductively through iterative refinement: (1) independent preliminary coding of responses, (2) comparison and discussion to develop unified codes, (3) refinement of code definitions, and (4) systematic application to all responses. Any coding discrepancies were resolved through consensus discussion involving the two coders and principal investigator until agreement was reached for each response[42,43]. Twenty-seven codes were ultimately developed. We retained themes describing food quality, variety, and affordability; chain versus independent store presence; transportation and geographic barriers; and structural factors influencing these conditions, as these related directly to our quantitative research questions.”

 Inter-coder reliability statistics were not calculated as all coding decisions were reached through consensus, ensuring consistent application while allowing nuanced interpretation.

Introduction
6. The background is conceptually well grounded but occasionally slides into causal rhetoric and cites pre-pandemic sources without explaining why pandemic-era mobility shifts warrant a fresh investigation.

 Re-framing statements to emphasize associations and integrating more recent “food apartheid” literature would sharpen the rationale.

Response: We reviewed the introduction to ensure that causal language was not used. We added a sentence explaining why pandemic-era mobility shifts warrant fresh investigation of food access patterns “The pandemic's impact on transportation patterns, particularly reduced public transit usage and increased reliance on personal vehicles, may have differentially affected access to chain versus independent stores. “ (Line 139-141).

We strengthened the discussion of food apartheid in lines 97-106: “Unlike "food desert," which can imply a naturally occurring phenomenon, "food apartheid" correctly frames the issue as a human-created system of segregation which led to an inadequate, inequitable, and unjust food environment[19]. This framework aligns with conceptualizations of structural racism as "the totality of ways, in which societies foster racial discrimination, via mutually reinforcing inequitable systems" including housing, employment, and other sectors that shape neighborhood environments[20]. The built environment becomes a key mechanism through which these mutually reinforcing systems operate, as zoning policies, transportation infrastructure, and retail development decisions systematically determine which communities have access to quality food retail and which do not[21,22].”

Results and Interpretation
7. Quantitative findings are reported with log-transformed coefficients that cannot be translated into daily intake without back-transformation; odds ratios for food-security status sometimes lack 95 % confidence intervals in text and tables. Qualitative themes appear as a standalone paragraph rather than being woven into the discussion, forfeiting the explanatory power that mixed methods can offer. Effect sizes may be inflated by the missing confounders noted above; re-specifying the models and providing variance-inflation factors would clarify robustness.

Response: We back-transformed log coefficients for fruit and vegetable intake for interpretability and ensured all confidence intervals are reported consistently, see abstract line 28 “1.18 times more per day,” and results lines 297-302: “Coefficients were back-transformed from the log scale to provide interpretable effect sizes. Chain store shoppers had significantly higher fruit and vegetable intake com-pared to independent store shoppers across all models: Model 1 1.28 times per day, (95% CI: 1.11, 1.48, p<0.001); Model 2 1.26 times per day, 95% CI: 1.09, 1.46, p=0.002); Model 3 1.18 times per day, 95% CI: 1.01, 1.38, p=0.037); and Model 4 (1.18 times per day, 95% CI: 1.01, 1.38, p=0.042).” See above our response to point #3 regarding specifying models.   We have added VIF analyses for all models, see lines 214-247: “To assess the presence of multicollinearity, variance inflation factors (VIF) for all variables were determined. Variance inflation factors for these models are found in Supplemental Tables 2-5. All values were below conventional thresholds (all VIF < 10), indicating no problematic multicollinearity [42]” and Supplemental Tables 2-5.

We have now integrated qualitative findings throughout the discussion rather than presenting them as standalone paragraphs, as suggested. See lines 445-456, 460-462, Lines 468-47, and lines 493-497.

Reviewer 3 Report

Comments and Suggestions for Authors

The manuscript has a good structure, but the following comments should be addressed before resubmitting it.

1) The study needs to be improved by anchoring its inquiry in a more robust theoretical framework, such as the Social Determinants of Health (SDOH), Ecological Systems Theory, or Critical Race Theory. In my opinion, incorporating one or more of these frameworks might enhance the study and fortify the understanding of systemic injustices.

2) As the COVID period is already passed, I suggest the author consider replacing this statement with another one; e.g., pandemic.

3) The study acknowledges the cross-sectional design, but a more rigorous discussion of its implications for causality is necessary. The readers should infer causality between grocery type and diet or food security, which currently is not clear in the Discussion and Conclusion of this study.

4) In the Methods section (recruitment and sampling), it was mentioned that reliance on online surveys and social media advertisements likely excluded key populations without digital access. But there is not any clear reason for this; please consider adding a more critical discussion on this.

5) Categorizing stores as simply “chain” vs. “independent” misses important heterogeneity within each group (e.g., food cooperatives, ethnic markets). It is important that you can discuss how this simplification might mask nuances in food access and quality.

6) The authors mentioned the use of the DSQ; however, this should be discussed more critically, as this screener is known to have low sensitivity for dietary variety and portion size, which may underestimate true dietary inadequacies in food-insecure populations.

7) Although food affordability is mentioned by a number of participants, in this study, more general inflation patterns in food costs between 2021 and 2022 are ignored.

8) This study discusses income and gender separately, but it doesn't delve into the intersectionality of these variables. In fact, intersectional analysis can better reveal the more complex barriers faced by, for example, low-income women.

9) While recommendations such as “support local retailers” and “enhance public transit” are valid, they are too general; please consider referencing specific, evidence-based programs (e.g., Double Up Food Bucks in Michigan) and their effectiveness.

10) The study mentioned “food apartheid," which needs more precise engagement with its meaning and implications, ideally with stronger academic citation support.

Author Response

Thank you for your valuable suggestions on theoretical frameworks and the importance of addressing systemic inequities in food access research. Your recommendations to incorporate SDOH perspectives and provide more specific policy recommendations have helped us strengthen this manuscript. 

Author Response : 

Reviewer 3

Thank you for your valuable suggestions on theoretical frameworks and the importance of addressing systemic inequities in food access research. Your recommendations to incorporate SDOH perspectives and provide more specific policy recommendations have helped us strengthen this manuscript.

  • The study needs to be improved by anchoring its inquiry in a more robust theoretical framework, such as the Social Determinants of Health (SDOH), Ecological Systems Theory, or Critical Race Theory. In my opinion, incorporating one or more of these frameworks might enhance the study and fortify the understanding of systemic injustices.

Response: Thank you for this valuable suggestion. We incorporated a discussion of SDOH as follows: Lines 51-53 “As such, food insecurity is recognized as a social determinant of health (SDOH), which the Centers for Disease Control and Prevention define as “nonmedical factors that influence health outcomes[6].”; Lines 61-62 “In fact, the SDOH framework includes neighborhood and built environment as one of the five domains that shape health [8].”, and lines 142-147: “Our study investigates how food insecurity intersects with multiple SDOH do-mains, particularly through economic stability constraints (food insecurity) and neighborhood and built environment factors (food retail access, transportation usage). Understanding these interconnections helps contextualize how transportation barriers, store accessibility, and socioeconomic factors work together to create systematic dis-advantages in food access.”

  • As the COVID period is already passed, I suggest the author consider replacing this statement with another one; e.g., pandemic.

Response: We revised terminology from "COVID period" to "COVID-19 pandemic" throughout for clarity.

  • The study acknowledges the cross-sectional design, but a more rigorous discussion of its implications for causality is necessary. The readers should infer causality between grocery type and diet or food security, which currently is not clear in the Discussion and Conclusion of this study.

Response: We strengthened our discussion of cross-sectional design limitations regarding causality in lines 577-581: “The cross-sectional design does not permit examining causal relationships between food grocery store type and transportation, fruit and vegetable intake, and food insecurity. Future studies could use designs such as natural experiments or randomized controlled trials to better understand if there is a causal relationship between these variables, and if so, in which direction.”

  • In the Methods section (recruitment and sampling), it was mentioned that reliance on online surveys and social media advertisements likely excluded key populations without digital access. But there is not any clear reason for this; please consider adding a more critical discussion on this.

Response: We address the digital divide and its potential impact on sample representativeness in (Lines 582-588): “Another limitation is the potential for non-response bias, as the response to the mailed flyers was low and the use of social media advertisements may have introduced selection bias by reaching a more technologically engaged subset of the population. This recruitment approach likely excluded the most economically disadvantaged residents, as it has been reported that 13% of adults with household incomes below $30,000 lack access to internet technologies at home[70]”

  • Categorizing stores as simply “chain” vs. “independent” misses important heterogeneity within each group (e.g., food cooperatives, ethnic markets). It is important that you can discuss how this simplification might mask nuances in food access and quality.

Response: We added discussion acknowledging the heterogeneity within chain and independent store categories in Lines 403-407: “While our binary store categorization enabled analysis of broad retail patterns, it obscures important heterogeneity within store types, including ethnic markets, food co-operatives, and varying price points that may independently influence food access and dietary outcomes among diverse populations”

  • The authors mentioned the use of the DSQ; however, this should be discussed more critically, as this screener is known to have low sensitivity for dietary variety and portion size, which may underestimate true dietary inadequacies in food-insecure populations.

Response We have added a discussion of DSQ limitations in Line 591-595: “Additionally, the dietary assessment used in this study, the DSQ, has limited sensitivity for dietary variety and portion size, and is known to typically result in an underestimate of fruit and vegetable intake. However, it can capture meaningful directional differences in fruit and vegetable consumption patterns when comparing intake among two groups [55]. 

  • Although food affordability is mentioned by a number of participants, in this study, more general inflation patterns in food costs between 2021 and 2022 are ignored.

Response: We incorporated Detroit-specific food inflation data showing 3% higher costs compared to national averages during our study period (Line 411-418). “The study period coincided with unprecedented food price inflation, making Detroit a particularly critical location for examining food shopping behaviors. The pandemic related global market disturbances resulted in the largest and most rapid increase in food inflation in the United States in over 40 years[46]. The Detroit-Warren-Dearborn metropolitan statistical area experienced disproportionately severe food price increases, with the consumer price index for foods consumed at home rising 10.0% in December 2021, 11.4% in February 2022, 14.3% in April 2022, and 16.6% in June 2022, consistently exceeding national averages by approximately 3%” This context is woven into both the introduction and discussion to highlight the importance of studying food access during this period.

Mention in Introduction and Discussion

  • This study discusses income and gender separately, but it doesn't delve into the intersectionality of these variables. In fact, intersectional analysis can better reveal the more complex barriers faced by, for example, low-income women.

Response: We appreciate this point, and have added to lines 598-602, “Lastly, as this study focused on the general population of Detroit, it provides a broad overview of the relationships examined, but does not explore the experiences of population subgroups. Future research is needed that focuses on the experiences of populations that are most affected by food insecurity, such as low-income women with children.

  • While recommendations such as “support local retailers” and “enhance public transit” are valid, they are too general; please consider referencing specific, evidence-based programs (e.g., Double Up Food Bucks in Michigan) and their effectiveness.

Response: We have included discussion of DUFB, including citations on its effectiveness in lines (Line 505-508) “ Incentive programs like Double Up Food Bucks, which matches SNAP benefits dollar-for-dollar when purchasing fruits and vegetables at participating retailers, have demonstrated effectiveness in increasing fruit and vegetable consumption among low-income households. [63–65],” and have made other small modifications to this paragraph.

  • The study mentioned “food apartheid," which needs more precise engagement with its meaning and implications, ideally with stronger academic citation support.

Response: We strengthened the discussion of food apartheid in lines 97-106: “Unlike "food desert," which can imply a naturally occurring phenomenon, "food apartheid" correctly frames the issue as a human-created system of segregation which led to an inadequate, inequitable, and unjust food environment[19]. This framework aligns with conceptualizations of structural racism as "the totality of ways, in which societies foster racial discrimination, via mutually reinforcing inequitable systems" including housing, employment, and other sectors that shape neighborhood environments[20]. The built environment becomes a key mechanism through which these mutually reinforcing systems operate, as zoning policies, transportation infrastructure, and retail development decisions systematically determine which communities have access to quality food retail and which do not[21,22].”

Round 2

Reviewer 2 Report

Comments and Suggestions for Authors

The paper reports a cross-sectional online survey (December 2021 – May 2022) of 656 adult Detroit residents that links the type of grocery store used (chain vs independent) to transportation mode, food-security status and self-reported fruit-and-vegetable intake. Multivariable logistic and linear models—with four tiers of adjustment culminating in age, sex, race, income-to-poverty ratio and SNAP participation—show that having a personal vehicle predicts shopping at chain stores (OR ≈ 1.9) and that doing so is associated with ~1.18 extra fruit-and-vegetable servings per day after covariate control. Qualitative thematic analysis of two open-ended questions identifies three recurring barriers—food quality/availability, accessibility (especially transport), and affordability—that deepen the quantitative findings.

Appraisal of the authors’ revisions

  • Causal wording toned down – Throughout the abstract and main text, phrases such as “increased the odds” were replaced by “was associated with,” aligning claims with a cross-sectional design.
  • Key survey logistics added – Field dates now appear in the abstract and the split between flyer (n = 219) and Facebook (n = 437) respondents is reported, improving transparency; however, invitation counts and ad impressions remain undocumented, so overall response rate is still unknowable.
  • Sample representativeness discussed – A new comparison with Detroit census data and a limitation paragraph on non-response bias candidly describe the skew toward older, more educated, lower-income women; this acknowledgement is welcome.
  • Model specification strengthened – SNAP participation has been added to create a fourth adjustment model and authors now report variance-inflation factors (< 10) to rule out collinearity.
  • Remaining omitted variables – Distance to the usual store and a price proxy are still unavailable; authors correctly flag this as a limitation but the absence could bias coefficients if economic accessibility drives store choice.
  • Store classification clarified – A reproducible definition for “chain” vs “independent” (full-line, corporate vs locally owned) plus consensus resolution of ambiguous cases removes earlier ambiguities.
  • Qualitative rigour improved but can go further – The iterative codebook process is described; consensus coding in lieu of intercoder reliability is defensible, yet even a brief report of initial agreement (e.g., % overlap before reconciliation) would bolster trustworthiness.
  • Integration of qualitative insights – Themes are now woven through the discussion, illustrating how transport barriers and perceived food quality contextualise statistical results—a clear advance over the standalone paragraph criticised earlier.
  • Abstract clarity – Back-transformed dietary effect sizes are now expressed in “times per day,” but converting this to standard cup-equivalents (or at least clarifying it reflects ~1.2 servings) would aid public-health readers.
  • Limitations section expanded – Cross-sectional caveats, missing price data, potential selection bias from online recruitment and DSQ measurement error are now spelt out, giving a balanced tone.
  • Overall response quality – The authors respond point-by-point, adopt nearly all substantive recommendations and document line changes, reflecting constructive engagement with review.

The revision materially improves clarity, methodological transparency and alignment between evidence and claims. Remaining gaps—chiefly unresolved exposure mis-measurement (price/distance) and limited reliability metrics for qualitative coding—are noted honestly but could still influence inference; consider whether these caveats warrant further minor edits before acceptance.

Suggested minor edits before final acceptance

  • Eliminate a duplicated sentence in the limitations. Lines 582 and 588 of the PDF each begin “Another limitation is the potential for non-response bias”; delete the second instance to avoid redundancy.
  • Standardise the dietary effect unit in the abstract. Replace “1.18 times more per day” with a commonly understood metric such as “≈ +1.2 cup-equivalents of fruit + vegetables per day” so readers need not convert “times/day.”
  • Report (or explicitly acknowledge the absence of) recruitment denominators. The Methods now state that 10 000 households received flyers and give the split of completed surveys, yet Facebook ad impressions and flyer-return counts remain unknown. Add one line clarifying that these figures were not tracked, so a conventional response rate cannot be computed.
  • Clarify the “speed check” criterion. Justify or cite a source for defining “impossibly fast completion” as “< 30 % of the median survey time” to head off reviewer questions about data-quality thresholds.
  • Add a brief intercoder-agreement indicator. Even a single statistic (e.g., “initial code overlap = 82 % before consensus”) would bolster the transparency of the qualitative analysis without requiring extra analysis.
  • Insert the IRB protocol number. The ethics statement confirms approval but omits the reference ID; adding it is routine for many journals.
  • Harmonise p-value formatting. Throughout the manuscript, use one style—preferably “p < 0.001” with a leading zero and non-breaking space—to replace variants such as “p<.001” or “p <0.001.”
  • Tighten author contribution and funding sections. Delete repeated phrasing (“Another limitation…” appears twice) and ensure each funding source is listed only once for a cleaner close.

Each fix is editorial rather than substantive, so the authors can implement them quickly without further analysis.

Author Response

Thank you again for your careful review of our paper. We believe that your contributions have helped to improve our paper.

Suggested minor edits before final acceptance

  1. Eliminate a duplicated sentence in the limitations. Lines 582 and 588 of the PDF each begin “Another limitation is the potential for non-response bias”; delete the second instance to avoid redundancy.

Response: Thank you for pointing out the duplicate, we have deleted the duplicated sentence.  

  1. Standardise the dietary effect unit in the abstract. Replace “1.18 times more per day” with a commonly understood metric such as “≈ +1.2 cup-equivalents of fruit + vegetables per day” so readers need not convert “times/day.”

Response: We agree that servings or cups is a more commonly understood metric.  However, to stay as true as possible to what the dietary assessment instrument measures, which is frequency and not quantity, we are retaining times per day as the unit of presenting fruit and vegetable intake.

  1. Report (or explicitly acknowledge the absence of) recruitment denominators. The Methods now state that 10 000 households received flyers and give the split of completed surveys, yet Facebook ad impressions and flyer-return counts remain unknown. Add one line clarifying that these figures were not tracked, so a conventional response rate cannot be computed.

Response: We added a clarifying line (167)We were not able to calculate survey response rate because neither the number of flyers distributed by food assistance organizations nor the Facebook ad impressions were tracked.

  1. Clarify the “speed check” criterion. Justify or cite a source for defining “impossibly fast completion” as “< 30 % of the median survey time” to head off reviewer questions about data-quality thresholds.

Response: We apologize, the threshold was 50% of the median not 30%, and we have added the relevant citation.

  1. Add a brief intercoder-agreement indicator. Even a single statistic (e.g., “initial code overlap = 82 % before consensus”) would bolster the transparency of the qualitative analysis without requiring extra analysis.

Response: We added “Prior to consensus 63% of responses were coded identically, an additional 20% had at least one code applied by both coders, 17% had no overlapping codes.”  At line 228-230).

  1. Insert the IRB protocol number. The ethics statement confirms approval but omits the reference ID; adding it is routine for many journals.

Response: We have added the IRB protocol number IRB-21-04-3429

  1. Harmonise p-value formatting. Throughout the manuscript, use one style—preferably “p < 0.001” with a leading zero and non-breaking space—to replace variants such as “p<.001” or “p <0.001.”

Response: Thank you for pointing this out. The leading zeroes and non-breaking spaces have been changed throughout the paper.

  1. Tighten author contribution and funding sections. Delete repeated phrasing (“Another limitation…” appears twice) and ensure each funding source is listed only once for a cleaner close.

Response: Thank you for pointing out the repeated phrasing and have deleted it. We have also tightened the author contribution section.

Reviewer 3 Report

Comments and Suggestions for Authors

Accept in present form

Author Response

Thanks again for your time and expertise reviewing this paper.